# A Metabolomics Investigation of the Metabolic Changes of Raji B Lymphoma Cells Undergoing Apoptosis Induced by Zinc Ions

**DOI:** 10.3390/metabo11100689

**Published:** 2021-10-07

**Authors:** Naeun Yoon, Hyunbeom Lee, Geonhee Lee, Eun Hye Kim, Seong Hwan Kim, Jeong-O Lee, Yunseon Song, Jinyoung Park, So-Dam Kim, Yeojin Kim, Byung Hwa Jung

**Affiliations:** 1Molecular Recognition Research Center, Korea Institute of Science and Technology, Seoul 02792, Korea; youne@sm.ac.kr (N.Y.); hyunbeom@kist.re.kr (H.L.); jypark@kist.re.kr (J.P.); 2College of Pharmacy, Sookmyung Women’s University, Seoul 04310, Korea; yssong@sm.ac.kr (Y.S.); 1531313@sookmyung.ac.kr (S.-D.K.); yojin92@naver.com (Y.K.); 3Advanced Materials Division, Korea Research Institute of Chemical Technology, Daejeon 34114, Korea; geonhl@krict.re.kr (G.L.); jolee@krick.re.kr (J.-O.L.); 4Drug Discovery Platform Research Center, Department of Drug Discovery, Korea Research Institute of Chemical Technology, Daejeon 34114, Korea; gracekim@krict.re.kr (E.H.K.); hwan@krict.re.kr (S.H.K.); 5Division of Bio-Medical Science and Technology, KIST School, Korea University of Science and Technology (UST), Seoul 02792, Korea

**Keywords:** cell metabolomics, metabolomics, cancer, zinc ion, apoptosis

## Abstract

Zinc plays a pivotal role in the function of cells and can induce apoptosis in various cancer cells, including Raji B lymphoma. However, the metabolic mechanism of Zn-induced apoptosis in Raji cells has not been explored. In this study, we performed global metabolic profiling using UPLC−Orbitrap−MS to assess the apoptosis of Raji cells induced by Zn ions released from ZnO nanorods. Multivariate analysis and database searches identified altered metabolites. Furthermore, the differences in the phosphorylation of 1380 proteins were also evaluated by Full Moon kinase array to discover the protein associated Zn−induced apoptosis. From the results, a prominent increase in glycerophosphocholine and fatty acids was observed after Zn ion treatment, but only arachidonic acid was shown to induce apoptosis. The kinase array revealed that the phosphorylation of p53, GTPase activation protein, CaMK2a, PPAR−γ, and PLA−2 was changed. From the pathway analysis, metabolic changes showed earlier onset than protein signaling, which were related to choline metabolism. LC−MS analysis was used to quantify the intracellular choline concentration, which decreased after Zn treatment, which may be related to the choline consumption required to produce choline-containing metabolites. Overall, we found that choline metabolism plays an important role in Zn-induced Raji cell apoptosis.

## 1. Introduction

Zn is an essential trace element that plays an important role in cell functioning as a nutrient, cofactor of enzymes, and intracellular mediator [1]. The role of zinc is known to be diverse, complex, and cell specific. Because of this importance, Zn in cancer has also been well studied and highlighted [2]. It has been reported that Zn induces apoptosis in many cancer cell types, including non-small-cell lung cancer (NSCLC), prostate cancer, and lymphoma [3,4,5]. Zn can exert the opposite effects, depending on the concentration, in human Burkitt lymphoma B cells. Higher concentrations of Zn activate a cell death pathway associated with caspase−3 activation, but a low concentration rather inhibits apoptosis [5]. The anticancer activity of ZnO in Raji B lymphoma has also been reported. ZnO thin film coating chips have anticancer activity based on their sustained release of Zn ions that downregulate antiapoptotic molecules and increase cellular oxidative stress [6].

Metabolomics can be used to identify metabolites that can present the phenotype of a cell or an organism. The possibility of discovering metabolites that affect phenotypes and their applications in metabolomics has been suggested [7]. Cell metabolomics can also provide the biomarkers of the pathogenesis and small molecular composition of apoptotic cells. Investigation of the metabolic mechanism under apoptosis can provide insight into understanding disease and drug discovery. Thus, several biomarkers of apoptosis have been discovered in cancer cells [8]. A decrease in choline, taurine, and glutathione in apoptotic HL60 leukemia cells, as well as a decrease in glucose uptake in SKOV−3 and OVCAR-3 cells, have previously been reported [9,10]. However, the metabolic mechanism of Zn ion-induced apoptosis in lymphoma cells has not been fully examined. To address this, we aimed to discover the metabolic mechanism associated with Zn-induced apoptosis in Raji B lymphoma cells with metabolomics using ultra-high-performance liquid chromatography (UPLC) orbitrap–mass spectrometry (MS). Since Zn ions released from ZnO successfully lead to apoptosis in Raji cells [6], we used Raji B lymphoma cells treated with Zn ions from ZnO nanorods to study the metabolic mechanism of apoptosis.

Zn ions are also essential for the action of many enzymes, including DNA and RNA polymerases, which are involved in DNA damage and repair [11,12,13,14], along with the tumor suppressor protein p53 and metalloproteinases [15]. Zn ions can bind to some zinc finger domains (ZnF) in DNA and RNA polymerase, including DNA polymerase-α [16,17]. Therefore, it is necessary to understand the apoptosis effect of Zn at the protein level, as well as metabolic changes; thus, we explored protein phosphorylation to understand its endogenous alterations under Zn ion treatment. Finally, considering the cell metabolic profiles obtained, we suggest that Zn ion-induced metabolic effects can contribute to cell death.

## 2. Results

### 2.1. Identification of the Metabolic Changes by Metabolic Profiling after Zn Ion Treatment of Raji Cells 

First, we confirmed the Zn ion−induced apoptosis of Raji cells via flow cytometry (Figure 1), and untargeted metabolomics was used to profile the metabolic changes after apoptosis. The pooled QC samples were consistent in the principal component analysis (PCA) plots created during analysis, and the stability of the experiment was demonstrated (Appendix A).

The PCA score plots indicated a clear distinction between the Zn ion-treated and control groups (Figure 2). The R2 and Q2 values of the positive mode were R2X [1] = 0.691, R2X [2] = 0.767, Q2 [1] = 0.650, and Q2 [2] = 0.698, respectively. The negative mode values were R2X [1] = 0.438, R2X [2] = 0.612, Q2 [1] = 0.386, and Q2 [2] = 0.538, respectively. Thus, endogenous metabolism changes after Zn ion treatment are critical.

Figure 3 shows a heat map of 68 metabolites that were significantly altered after Zn ion treatment (Figure 3). The identified metabolites are described in Appendix A, and the related information (MSI level, peak retention time, and ionization mode) is also noted in Appendix A. The significantly changed metabolites had *p*-values less than 0.05 and *q*-values for the false discovery rate (FDR) less than 0.05. Among these metabolites, the levels of the 59, 58, and 51 metabolites changed significantly after Zn ion treatment for 12, 18, and 24 h, respectively.

Generally, the levels of amino acids and their analogs, organonitrogen and organooxygen compounds, fatty acyls, glycerophospholipids, nucleosides, and nucleotide classes showed opposite trends between the control and Zn ion−treated groups.

Prominent changes appeared in the levels of glycerophospholipids and fatty acyls. The levels of GPC, AA, DHA, DPA, and adrenic acid increased after Zn ion treatment for 12 h by 1100-, 80-, 43-, 46-, and 37-fold, respectively (Appendix A). The increasing pattern in the levels of these metabolites was maintained for 24 h, but this fold increase was reduced over time. The levels of phosphatidylcholine (0:0/18:0) and phosphatidylcholine (36:4) also increased over 24 h from 1−10−fold. However, the levels of most other fatty acyls and glycerophospholipids increased at 12 and 18 h, but decreased at 24 h.

Most amino acids, except for threonic acid and *n*−acetyl−dl−methionine, increased after 12 h of Zn ion treatment, but decreased at 18 h. On the contrary, all of the levels of threonic acid increased, but the level of n−acetyl−dl−methionine decreased 24 h after Zn ion treatment. In particular, the levels of fatty acyls, nucleotides, and nucleosides were markedly decreased 24 h after treatment.

### 2.2. Protein Phosphorylation Analysis

We analyzed 1380 proteins through a Full Moon kinase array to investigate the effect of Zn ions on the protein level. To determine the phosphorylated changes of each protein, the phosphorylated ratio was calculated as the phosphorylated−form protein intensity/unphosphorylated−form protein intensity. The top 10 highest and lowest changed phosphorylated/unphosphorylated protein ratios at each time point are shown in a heat map (Figure 4). Consistent with previous studies, the phosphorylation ratio of p53 (p−Ser392) decreased. The phosphorylation of p53 (Ser392) decreased the most 1 h after Zn ion treatment, with a ratio of 0.617. GTPase−activating protein (Ser387) exhibited the greatest increase in phosphorylation after Zn ion treatment, with a ratio of 3.806 at 24 h. The phosphorylation of PPAR−γ (Ser112) typically increased 10 min, 1 h, 6 h, and 12 h after Zn ion treatment, but decreased at 24 h. Phosphorylation of PPAR−γ at the Ser112 residue has been reported to decrease the transcriptional activity of PPAR-γ [18]. Similarly, the ratio (Zn ion−treatment/control) of the unphosphorylated PPAR−γ decreased to 0.8, 0.67, 0.63, and 0.448 from 10 min to 12 h after Zn ion treatment, respectively, but increased to 1.26 at 24 h. In addition, phosphorylated c−PLA2, which is required for arachidonic acid release, also increased 1.39−fold by 12 h after Zn ion treatment, and this result is consistent with the metabolomics results showing an increase in arachidonic acid after 12 h (Appendix A).

### 2.3. Ingenuity Pathway Analysis (IPA) and Metabolic Pathway Analysis by KEGG

IPA was performed to determine the relevant signaling networks with the identified metabolites and proteins. Cell death was upregulated, while cell survival and cell viability were decreased functionally 12 and 18 h after Zn ion treatment according to the “disease and function analysis” in the IPA program (Table 1, Figure 5). The activation z-scores for the functions of necrosis, the apoptosis of tumor cell lines, and the cell death of epithelial cells were altered more after 12 h of Zn ion treatment than after 18 h (Table 1). In particular, the metabolites associated with tumor cell death, namely, l-glutamic acid, arachidonic acid, stearic acid, acetyl−l−carnitine, and uridine, were clearly changed after 12 h of Zn ion treatment (Figure 5).

IPA was also performed with protein array data, and the canonical pathways are shown in Table 2. The change in canonical pathways for the proteins was prominent from 12 to 24 h. The five most significant canonical pathways altered by Zn ions were glioma signaling, endothelin−1 signaling, Fc epsilon RI signaling, phospholipase C signaling, and NF−κB signaling, which all increased over 12 h and then dramatically decreased at 24 h.

Comparing the disease and function results from the IPA analysis between the metabolite and the protein phosphorylation arrays, the metabolites related to cell death started to change between 12 and 18 h, whereas the proteins related to cell death were dramatically changed at 24 h. The activation z−score of the viability of the tumor cell lines was 3.07 at 12 h and decreased to −3.59 at 24 h in the protein analysis. However, the metabolomics data began to show a decrease, to −2.22, at 12 h. Therefore, it is suggested that the Zn ion-induced metabolic changes in apoptosis precede changes in relevant proteins.

According to the KEGG database, the identified metabolites are mostly involved in choline metabolism in cancer. Phospholipase C (PLC) signaling is also involved in this metabolic pathway, which was highly associated with the response to Zn ion treatment with IPA analysis (Table 2).

### 2.4. Choline Metabolism in Cancer Altered by Zn Ions

From the pathway analysis, it was found that the metabolites involved in choline metabolism were increased by zinc ion treatment (Figure 6A), and this pathway is known to be a target for antitumor therapy [19]. The levels of metabolites in this pathway increased 12 h after Zn ion treatment, and the levels returned to baseline at 24 h.

To determine whether Zn ions affect the intracellular choline concentration, we quantified the choline concentration in cells using LC−MS/MS. The choline level in the cells was significantly decreased by Zn ion treatment for the first 24 h, but the choline level differences between the control and treated groups were smaller at 24 h than at 12 h (Figure 6B). The level of choline decreased, but the levels of FAs and glycerophospholipids increased after Zn ion treatment. This finding indicates that the choline production of FAs and phospholipids was exhausted in the early phase of Zn ion treatment, which might lead to cell apoptosis.

### 2.5. Cell Survival with Fatty Acid Treatment

We found a significant increase in fatty acids and GPC at the same time that the concentration of choline decreased and thus predicted that this change would affect Zn ion-induced apoptosis. Therefore, we conducted cell survival experiments to test whether increasing fatty acyls and glycerophosphocholine (GPC) contributed to cancer cell death induced by Zn ions. Glycerophosphocholine (GPC), arachidonic acid (AA), docosahexaenoic acid (DHA), and docosapentaenoic acid (DPA), the levels of which were increased more than 50-fold by Zn ion treatment, were added to the cells.

The results showed that only AA significantly induced cell death. The survival rate after AA treatment decreased to 43% at 12 h and to 63% at 24 h (Figure 7A) in a dose-dependent manner (Figure 7B). The apoptosis rate was also determined by FACS analysis. Treatment of Raji cells with 20 μM of AA for 24 h increased early and late apoptosis by 21.4% and 16.2%, respectively (Appendix A).

Since Zn ion−induced apoptosis proved to be caspase-dependent apoptosis [9], we conducted an immunoblot assay on Raji cells to confirm that AA also induced caspase-dependent apoptosis. We determined the AA concentration for the immunoblot assay by referring to the endogenous AA concentration in Raji cells after 12 h of Zn ion treatment, since the AA area ratio was highest at 12 h after Zn treatment in metabolomics. The intracellular concentration of AA significantly increased (*p* < 0.05) in the 12 h of Zn ion treatment, and the concentrations were 0.12 ± 0.012 μM per cell (10^6^ cells) in the control and 13.98 ± 2.475 μM per cell (10^6^ cells) in the Zn 12 h treatment group. Considering this concentration, the concentration for the immunoblot assay was determined to be 10 μM.

We found that cleaved caspase −3, −9, and PARP were much more abundant in the AA-treated cells than in the control cells, and cleaved PARP was significantly upregulated 6 h after treatment (Figure 7C,D). These results indicate that an increase in intracellular AA levels can induced caspase-dependent apoptosis in Raji cells.

## 3. Discussion

Zn ions are required for DNA and RNA polymerase and tumor protein p53 to properly function, and are involved in repairing DNA damage and signals [11,12,13,14]. It has also been reported that Zn ions can induce oxidative stress and downregulate antiapoptotic molecules in Raji B lymphoma cells [6]. Herein, using a metabolomics approach, we demonstrated the metabolic mechanism of Zn ion-induced apoptosis of Raji cells. Based on cellular metabolic profiling with Zn ion−treated Raji B lymphoma cells, we showed that Zn ions caused a significant change in cellular metabolites. Interestingly, the levels of most metabolites were increased within 12 h of Zn ion treatment, while their levels diminished by 24 h. Among the metabolites, fatty acids and glycerophospholipids showed the most prominent level changes. Specially, GPC, AA, DHA, and DPA were drastically higher in the Zn−treated groups, and GPC has already been reported as a metabolite associated with apoptosis [20,21,22]. Moreover, the alteration of GPC in this study showed an initial and drastic increment (1100−fold) compared to any other research. However, GPC was not the metabolite contributing to apoptosis, so elevated GPC could be thought of as the result of cell membrane breakdown by apoptosis.

In particular, the level of arachidonic acid (AA) increased after Zn ion treatment. The enzyme c−PLA2 releases AA for eicosanoid synthesis [23], and it has been reported that the overexpression of c−PLA2 enhanced cell death, while the suppression of c−PLA2 decreased cell death [24]. c−PLA2 and AA both increased after 12 h after Zn treatment in this study. Notably, AA can induce the apoptosis via activation of caspase−3, and its addition leads to mitochondrial permeability transition, which in turn releases cytochrome c [25]. As previously reported, among the 20 fatty acids tested, AA has been reported to be a unique fatty acid that has the ability to induce the apoptosis of cancer cells [26], and AA has been investigated as a cause of cancer cell death [27,28]. In addition, AA has been reported to induce apoptosis in various cell types. Y79 cell exposure to AA revealed a significant increase in lipid peroxidation end products and a sharp drop in mitochondrial membrane potential, resulting in apoptosis [29]. AA inhibits SREBP−1 and blocks the endogenous synthesis of fatty acids, leading to endoplasmic reticulum (ER) stress and apoptosis in HT−29 colon cancer cells [30]. Moreover, treatment with apoptotic concentrations of AA involves changes in oxidative stress and eicosanoid biosynthesis in leucocytes [27]. We also confirmed that exogenous treatment with AA leads to the caspase-dependent apoptosis of Raji cells.

The levels of DHA and phosphatidylcholine also increased after Zn ion treatment, and the pattern of change was similar to that of AA. The anticancer effects of DHA are known to be due to the induction of ROS and subsequent peroxidation of lipids [31,32]. Phosphatidylcholine is also suggested to induce apoptosis in hepatic cancer cells and T3-L1 adipocytes through the activation of p38 and JNK and cleavage of caspase−8, −9, −3, and PARP [33].

Although not as dramatic as the metabolic change of the lipids, a change in the level of amino acids (from approximately 1− to 5−fold) was also observed with Zn ion treatment. In particular, an increase in glutamic acid and a decrease in uridine were found to contribute to cell death in the IPA analysis (Figure 5). Uridine has been previously reported to be deficient in cells undergoing apoptosis via the mitochondrial pathway [34,35].

Proteins play important roles in cell apoptosis; therefore, we used the protein Full Moon Phospho−Explorer Antibody Array to determine how proteins change after Zn ion treatment. Previously, a decrease in phosphorylated p53 was shown to lead to the apoptosis of SMMC−7721 and HuH−7 cells, and it decreased after Zn treatments [36]. PPAR−γ levels decreased until 12 h after Zn ion treatment, but PPAR-γ was activated at 24 h. The ligands of PPAR−γ are long−chain unsaturated and polyunsaturated fatty acids (PUFAs) and metabolites of these fatty acids [37,38]. Activation of this pathway induces apoptosis and suppresses the proliferation of cancer cells [39,40]. DHA can induce the expression of p53 protein in hematopoietic Reh cells in a PPAR−γ−dependent manner. Therefore, we assume that significant increases in fatty acyl induced by Zn ion treatment can contribute to apoptosis by decreasing the phosphorylation of PPAR−γ. However, this finding cannot be experimentally proven, since blocking PPAR−γ with its antagonist does not lead to a significant change in the cell survival rate (Appendix A).

From the IPA analysis, the five canonical pathways most significantly changed by Zn ions were glioma signaling, endothelin−1 signaling, Fc epsilon RI signaling, PLC signaling, and NF−κB signaling pathways (Table 1). PLC signaling was one of the most significantly changed pathways after Zn ion treatment. PLC plays a significant role in transmembrane signaling, as it catalyzes the release of arachidonic acid from phosphatidylinositol and phosphatidylcholine, and a zinc-sensing receptor triggers the activation of PLC [41,42].

According to the KEGG pathway search, the metabolites changed by Zn ion treatment are commonly found in the choline metabolism in cancer; therefore, we hypothesized that Zn ion treatment can affect the choline metabolism in Raji cells. The increase in phosphocholine (PC) and total choline−containing compounds has been identified as a metabolic characteristic of tumors [43,44].

To address the association between Zn ions and choline, we quantified the concentration of choline in the cells. The results showed that the choline level decreased upon Zn ion treatment up to 24 h, but the choline level differences between the control and treated groups were smaller at 24 h than at 12 h. On the contrary, the levels of FAs and glycerophospholipids increased within 24 h after Zn ion treatment, and the increase was higher at 12 h than at 24 h. Therefore, we hypothesize that the increases in FAs and glycerophospholipids are the result of choline consumption.

Finally, it was concluded that the metabolism of GPC and fatty acids was prominently changed in Zn ion−induced apoptosis. Metabolic changes were found to be initiated earlier than protein signaling, and cell apoptosis was proven to be closely related to the choline metabolism.

Therefore, it is thought that the choline metabolism is important in Zn ion−induced Raji cell apoptosis, but other metabolic pathways may also be involved in cell death.

## 4. Materials and Methods

### 4.1. Materials

The Raji cell lines were obtained from the American Type Culture Collection (Manassas, VA, USA). All the cell culture materials, including fetal bovine serum (FBS), antibiotic-antimycotic and medium, were purchased from Gibco BRL (Gaithersburg, MD, USA). LC−MS−grade methanol was purchased from Burdick & Jackson (SK Chemicals, Ulsan, Korea). Ultrapure water (18.2 MΩ∙cm) was obtained using a Milli−Q apparatus from Millipore (Milford, CT, USA). Formic acid, reserpine, choline chloride, arachidonic acid (AA), glycerophosphocholine (GPC), cis−7, −10, −13, −16, and −19−docosapentaenoic acid (DPA), and cis−4, −7, −10, −13, −16, and −19−docosahexaenoic acid (DHA) were all purchased from Sigma Aldrich (St. Louis, MO, USA). Phospho Explorer antibody microarrays designed and manufactured by Full Moon Biosystems, Inc. (Sunnyvale, CA, USA), contain 1318 antibodies. The 16:0−d_3_1−18:1 PC standard was obtained from Avanti Polar Lipids, Inc. (Alabaster, AL, USA). A CCK−8 assay kit was obtained from Sigma Aldrich (St. Louis, MO, USA). For Western blotting, horseradish peroxidase−conjugated anti−rabbit (#7074), horseradish peroxidase−conjugated anti−mouse (#7076), and anti−cleaved caspase 9 (#31245) antibodies were purchased from Cell Signaling Technology (Danvers, MA 01923, USA). Anti−cleaved caspase 3 (PA5−23921) and anti−cleaved PARP (44−698G) were purchased from Thermo Scientific (San Jose, CA, USA).

### 4.2. Cell Culture

Hexamethylenetetramine (HTMA) and 20 mM zinc nitrate were dissolved in 500 mL of distilled water using the hydrothermal growth method. ZnO rods were obtained in bulk (2 g) and dried in an oven at 30 °C for 24 h. The dried ZnO rods were suspended and sonicated in 500 mL of RPMI media with 10% FBS and 1% antibiotic−antimycotic for 4 h. ZnO rods were filtered, and the Zn ion concentration in the medium was measured by quantitative analysis of the elements using inductively coupled plasma atomic emission spectroscopy.

The Raji cells were cultured with RPMI−1640 containing 10% heat−inactivated fetal bovine serum and 1% antibiotics (100 U/mL penicillin and 100 µg/mL streptomycin) in a humidified atmosphere of 5% CO_2_ at 37 °C [6] in the presence and absence of Zn ions (10 mg/L) for 12, 18, and 24 h to compare the metabolic differences.

### 4.3. Metabolite Extraction

Metabolites were extracted from cells extracted as described in previous work [45]. The supernatant was transferred to a clean vial, and 10 μL of the supernatant was injected into an Ultimate 3000 UHPLC System-LTQ Orbitrap Velos Pro mass spectrometer (Thermo Scientific). A quality control (QC) sample was prepared by pooling equal volumes of each sample, and column conditioning was performed by injecting the QC sample 10 times before the analytical runs were performed [46]. The QC sample was also analyzed after each set of 10 analytical sample runs to evaluate the repeatability of the instrument.

### 4.4. Metabolomic Analysis

Cellular metabolic profiling was performed using an Ultimate 3000 UHPLC system consisting of an autosampler and a column oven coupled to an LTQ Orbitrap Velos Pro mass spectrometer (Thermo Scientific, San Jose, CA, USA). An ACQUITY UPLC HSS T3 column (2.1 × 100 mm, 1.8 μm; Waters, Milford, MA, USA) was used at flow rates of 0.4 mL min^–1^ and 40 °C.

Gradient elution was performed using mobile phase A (0.1% formic acid in distilled water) and mobile phase B (0.1% formic acid in methanol) [47]. All samples were analyzed randomly to eliminate potential effects of the analysis order. MS using an electrospray ionization (ESI) source was operated in both positive and negative ionization modes. The analyzed data were normalized by the DNA concentration of each sample. The DNA concentration of each sample was quantified using a Nano−MD (SINCO, Daejeon, Korea).

### 4.5. Kinase Arrays

For the Phospho Explorer array analysis, Raji cells were incubated under the same conditions as for the metabolomic analysis with Zn ions: at a concentration of 10 μg/mL and cultured for 10 min and 1, 6, 12, and 24 h. The culturing time was determined by considering that protein expression can change prior to a metabolic change [26]. A total of 1318 antibodies were used in the array to analyze protein phosphorylation expression and phosphorylated forms. A full Moon Phospho Explorer antibody array was performed using the protocols established by the manufacturer [48]. The cell lysate was biotinylated with an antibody array assay kit (Full Moon Biosystems, Sunnyvale, CA, USA). Antibody microarray slides were blocked to prevent the binding of slides and proteins throughout the entire coupling process. To prevent reagent coupling and aggregation, a coupling mixture of dry milk was used to block the antibody microarray slides, and thus, coupling was blocked. Each slide (containing six replicates) was hybridized, and Cy3 fluorescence was measured by a microarray scanner with a scan resolution of 10 mm (Agilent Technologies, CA, Santa Clara, CA, USA). The images were quantified using GenePix Pro software (version 7.0) (Agilent Technologies).

### 4.6. Data Preprocessing and Statistical Analysis for the Identification of Metabolites

The data were processed using Compound Discoverer 2.1 software (Thermo Scientific, San Jose, CA, USA) for peak deconvolution and data normalization. For the statistical analysis, the data were normalized to internal standards and DNA concentrations of each sample. Metabolic differences between groups were investigated by multivariate statistical analysis using the principal component analysis (PCA) algorithm in EZinfo (Umetrics software, Umea, Sweden). The quality of the PCA model was determined by cross-validation of parameters R2 and Q2, representing the explained variance and the predictive capability of the model, respectively [49]. Molecular identification and structure prediction were performed based on the exact mass retention time pairs using Xcalibur software (version 2.2) (Thermo Scientific, San Jose, CA, USA).

The identification of metabolites in the analysis was putatively based on annotated compounds corresponding to level 1 or 2 of the metabolomics standard initiative (MSI) [50]. The MS/MS spectral data were compared based on information from public databases and the interpretation of MS and MS/MS spectra. The identities of peaks were verified using an in-house database and online databases, including the Human Metabolome Database (http://www.hmdb.ca/ accessed on 15 January 2019), METLIN (http://metlin.scripps.edu/ accessed on 15 January 2019), ChemSpider (http://www.chemspider.com/ accessed on 22 January 2019), and KEGG (http://www.genome.jp/kegg/ accessed on 2 March 2019). Differences in the cellular levels of each metabolite in the heat map were statistically evaluated by Wilcoxon−Mann−Whitney tests using MetaboAnalyst (http://www.metaboanalyst.ca/ accessed on 26 July 2019) [51]. A Student’s *t*−test was used to test the altered metabolite features between the cells at different times after their treatment with Zn ions and the control. The Ingenuity Pathway Analysis (IPA) system (http://www.ingenuity.com/ accessed on 26 July 2019) was used to identify relevant pathways. A *p*-value of <0.05 was considered statistically significant: * *p* ≤ 0.05, ** *p* ≤ 0.01, *** *p* ≤ 0.001, and **** *p* ≤ 0.0001.

### 4.7. Quantitative Analysis of Choline in the Cells

To evaluate the metabolic change in the early phase of Zn ion treatment, the choline concentration in Zn ion-treated Raji cells was quantitatively analyzed. Metabolites in the control and 6, 12, and 24 h Zn ion−treated Raji cells were extracted as described for metabolic profiling, and 1−palmitoyl−d31−2−oleoyl−sn−glycero−3−phosphocholine (16:0–d31–18:1 PC) was used as the internal standard for the quantitative analysis of choline. The quantitative analysis was performed using an Exion LC chromatography system with triple-quadrupole mass spectrometry API4500 (AB Sciex) operated in positive ESI mode using multiple reaction monitoring. Chromatographic separation was performed using an Ultimate 3000 UHPLC system equipped with a UHPLC HILIC column (2.1 × 150 mm, 1.7 μm, Waters). Analyst software (version 1.5.2) (AB Sciex, Concord, ON, Canada) was used for data acquisition and to control the equipment.

### 4.8. Quantitative Analysis of Arachidonic Acid in the Cells

To clarify the AA concentration in Raji cells after Zn ion addition, we quantified the concentration of AA in the cell using Exion LC chromatography system with triple–quadrupole mass spectrometry API4500. The Raji cells were incubated with or without Zn ion treatment for 12 h. AA was extracted from Raji cell pellet as described in the metabolomics profiling. Chromatographic separation was assessed using ACQUITY UPLC BEH C18 column (2.1 × 100 mm, 1.7 µm, Waters, Milford, CT, USA), and reserpine (2 mg/L) was used as the internal standard. The intracellular concentration was calculated as the concentration from the standard curve. Calculated concentration of AA was normalized by the cell number (10^6^ cells).

### 4.9. Cell Viability Assay

To determine the cell survival response to metabolites, we performed a Cell Counting KIT−8 (CCK−8) cell viability assay. After being seeded in a 96−well microplate at a density of 1.5 × 10^5^ per well, the Raji cells were treated with arachidonic acid (AA), glycerophosphocholine (GPC), cis −7, −10, −13, −16, an −19−docosapentaenoic acid (DPA) in dimethyl sulfoxide (DMSO) at concentrations of 10 and 20 µM for 12 and 24 h. After incubation at 37 °C for an additional 4 h, the absorbance was measured at 450 nm by a microplate reader.

### 4.10. Western Blot Analysis

After the Raji cells were treated with 10 µM of AA, they were harvested and lysed using cell lysis buffer (Cell Signaling Tech, 9803S) following the manufacturer’s instructions to confirm the apoptosis induced by AA. Proteins in the lysates were separated by SDS−PAGE and transferred to nitrocellulose membranes by an iBlot 2 system (Thermo Fisher Scientific, San Jose, CA, USA). After blocking in 5% BSA in TBS containing 0.5% Tween −20, the membrane was probed with primary antibodies and detected with horseradish-conjugated antibodies. Chemiluminescence was measured and quantified using a Bio−Rad ChemiDoc imaging system (Bio−Rad, Seoul, Korea).

### 4.11. Annexin-V/Propidium Iodide Staining Assay

To identify the apoptotic Raji cells following Zn ion treatment, we performed an Annexin−V/propidium iodide (PI) staining assay using an EzWay Annexin−V−FITC apoptosis detection kit (K29100; KOMA Biotech, Seoul, Korea) in accordance with the manufacturer’s instructions. After treatment with 10 mg/L of Zn ions for 12, 18, and 24 h, the Raji cells were harvested (5 × 10^5^). Then, the cells were washed with cold PBS and resuspended in binding buffer containing PI and FITC−conjugated Annexin-V protein. The percentage of stained apoptotic cells was determined using flow cytometry (FACS) with a FACSCanto II cytometer. The percentage of the cells in the Annexin−V−FITC scatter plots was calculated for comparison.

## Figures and Tables

**Figure 1 metabolites-11-00689-f001:**
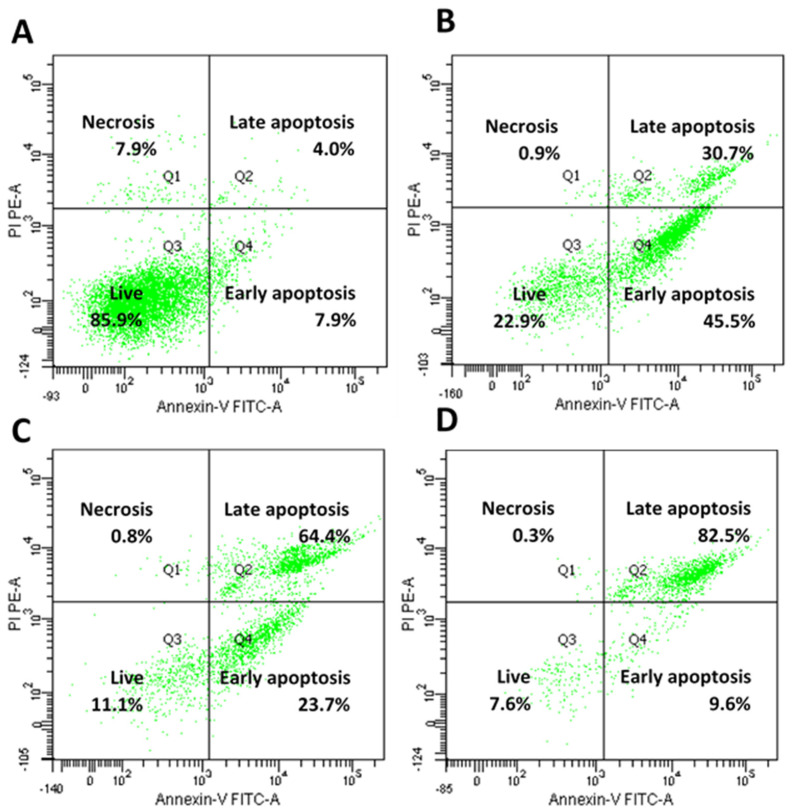
Apoptotic cell population evaluated by flow cytometry analysis following double staining with Annexin−V and propidium iodide (PI). (**A**) Raji cells without treatment; (**B**) Raji cells treated with Zn ion (10 mg/L) for 12 h, (**C**)18 h, and (**D**) 24 h.

**Figure 2 metabolites-11-00689-f002:**
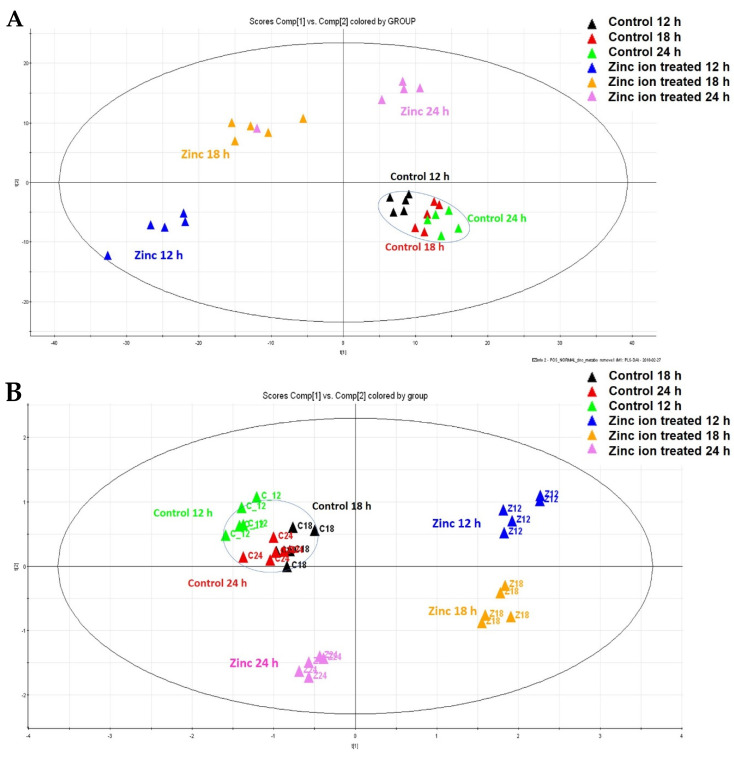
Multivariate statistical analysis based on the time−dependent metabolic profiling of Zn ion−treated Raji cells. (**A**) PCA score plot of the positive mode (R2X [1] = 0.691, R2X [2] = 0.767); (**B**) PCA score plot of negative mode (R2X [1] = 0.438, R2X [2] = 0.612). Green, control Raji cells incubated for 12 h; black, control Raji cells incubated for 18 h; red, control Raji cells incubated for 24 h; blue, Raji cells incubated with Zn ions for 12 h; yellow, Raji cells incubated with Zn ions for 18 h; pink, Raji cells incubated with Zn ions for 24 h (*n* = 5 in each group). All of the dots in the circle are controls.

**Figure 3 metabolites-11-00689-f003:**
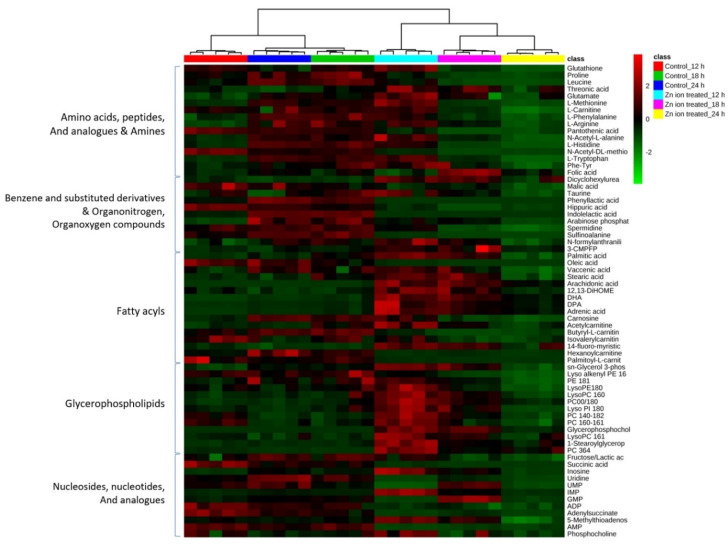
Heat map and cluster analysis of the metabolites discriminated after Zn ion treatment in the Raji cell lines. The distance measure was calculated by Euclidean distance, and the metabolites were verified by Student’s *t*−test. Significantly decreased metabolites are displayed in green, and increased metabolites are shown in red. Each group is indicated at the top of the figure. Red, control 12 h; green, control 18 h; blue, control 24 h; cyan, Zn ion treatment 12 h; pink: Zn ion treatment 18 h; yellow, Zn ion treatment 24 h (each *n* = 5).

**Figure 4 metabolites-11-00689-f004:**
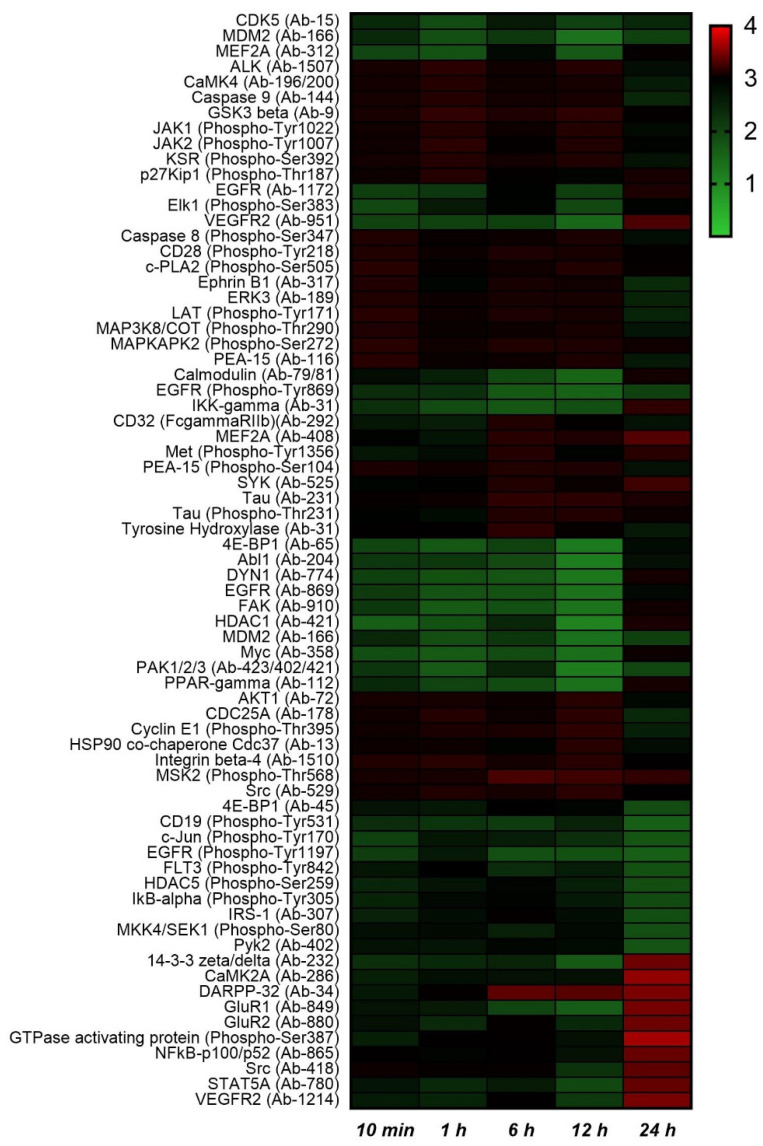
The top 10 highest and lowest changes in the phosphorylated/nonphosphorylated protein ratios induced by Zn ion treatment. The heat map value is the intensity ratio (intensity of phosphorylated protein/intensity of nonphosphorylated protein). The bottom line shows the time of Zn ion treatment. The red and green colors indicate increases and decreases, respectively.

**Figure 5 metabolites-11-00689-f005:**
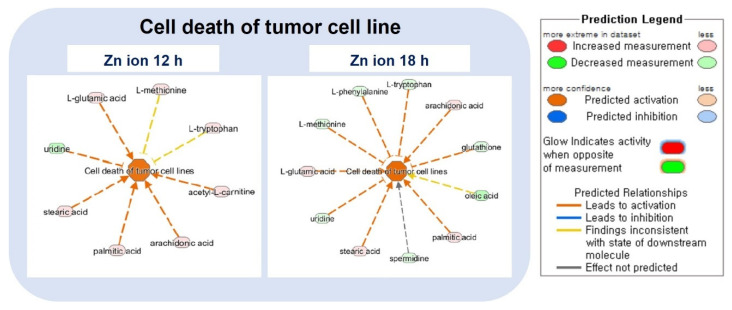
Top enriched diseases and functions by IPA analysis. Description of the cell death of the tumor cell line pathway in detail. The prediction legend is described in the box on the right.

**Figure 6 metabolites-11-00689-f006:**
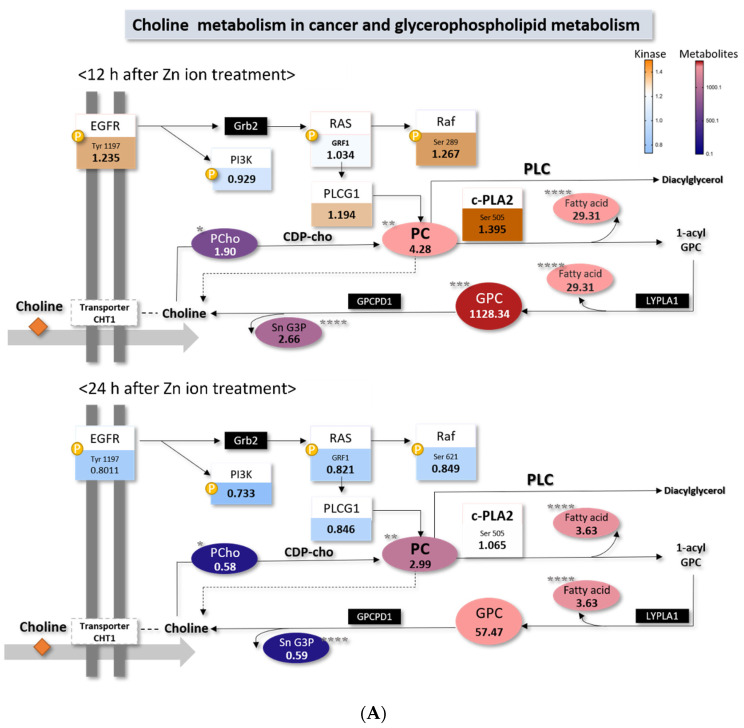
(**A**) Proposed choline metabolic pathway after Zn ion treatment. The number in the circle indicates the ratio of each metabolite between the control and Zn ion−treated groups (normalized intensity of Zn ion−treated sample/normalized intensity of control sample). The number in the box indicates the phosphorylated ratio of kinase (phosphorylated-form kinase intensity/unphosphorylated-form kinase intensity) between the Zn ion−treated groups divided by the control. The color of the red series indicates an increase in the ratio, and the blue series indicates a decrease in the ratio. Circles represent the metabolome; boxes represent kinases; ⓟ denotes phosphorylated kinase. Statistically significant: * *p* ≤ 0.05, ** *p* ≤ 0.01, *** *p* ≤ 0.001, and **** *p* ≤ 0.0001. (**B**) Choline concentration in Zn ion-treated Raji cells. The integrated peak intensities of choline were normalized by the concentration of DNA in each cell pellet. Cells were incubated in Zn ion−containing media (10 mg/L) for 6, 12, and 24 h (*n* = 3). * *p* < 0.05.

**Figure 7 metabolites-11-00689-f007:**
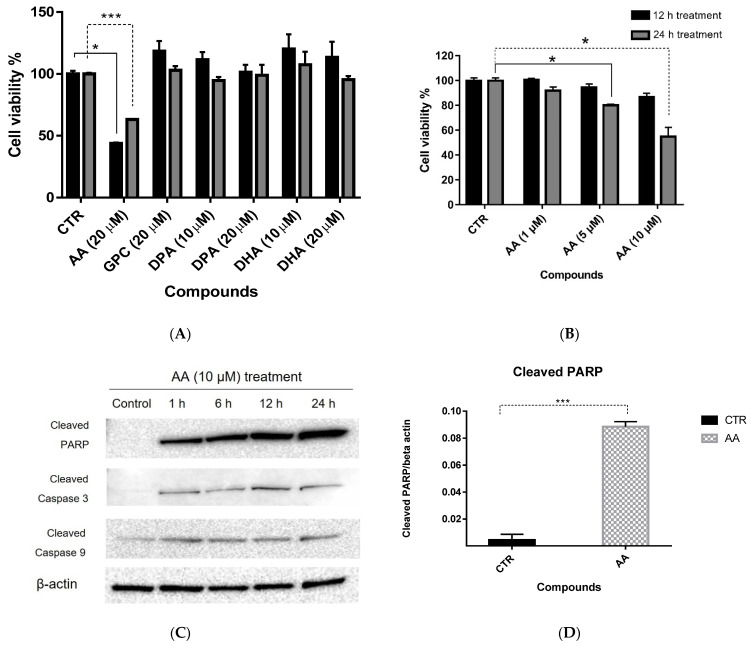
(**A**) Cell viability test after treatment of Raji cells with AA. AA, arachidonic acid (20 µM); GPC, glycerophosphocholine (20 µM); DPA, docosapentaenoic acid (10 µM). (**B**) Concentration-dependent AA treatment of Raji cells. The black bar represents the 12 h treatment, and the gray bar represents the 24 h treatment. (**C**) Western blot assays were performed on Raji cells to confirm that AA caused caspase-dependent cell death. Cells were treated with 10 µM of AA for 1, 6, 12, and 24 h. (**D**) Bar graphs show the quantification of cleaved PARP at 6 h after AA treatment. Cleaved PARP was significantly increased (each *n* = 3). Statistically significant: * *p* ≤ 0.05, *** *p* ≤ 0.001.

**Table 1 metabolites-11-00689-t001:** Top altered diseases and functional pathways compared to 12 and 18 h after Zn ion treatment.

Diseases and Biological Functions	Activation z-Score	
	Zn Ion 12 h	Zn Ion 18 h
Growth of organism	−2.88	−1.61
Cell death of tumor cell lines	2.63	1.83
Activation of microglia	1.98	1.98
Apoptosis of tumor cell lines	2.08	1.85
Killing of cells	1.96	1.94
Cell death of epithelial cells	2.40	1.43
Cell viability	−2.31	−1.47
Quantity of reactive oxygen species	2.12	1.63
Cell death of carcinoma cell lines	1.99	1.73
Cell viability of tumor cell lines	−2.22	−1.47
Cancer	−1.04	−2.58
Uptake of amino acids	1.57	−1.93
Cell survival	−2.18	−1.18

The activation z-score was calculated using the IPA software, which predicts whether a specific pathway is increased (positive z−score) or decreased (negative z−score).

**Table 2 metabolites-11-00689-t002:** Top canonical pathways and diseases and functions altered by Zn ion treatment from IPA analysis.

Pathway Name	Zn Ion Treatment Time (h)				
	0.16	1	6	12	24
Canonical Pathway					
Glioma Signaling	3.09	3.43	2.40	4.12	−4.46
Endothelin−1 Signaling	1.58	3.48	2.85	3.79	−5.06
Fc Epsilon RI Signaling	3.68	2.72	3.36	4.00	−4.00
Phospholipase C Signaling	2.99	1.91	4.63	3.81	−2.18
F−κB Signaling	3.64	2.83	1.75	4.18	−3.10
Leukocyte Extravasation Signaling	2.31	3.18	1.44	2.31	−4.91
CCR3 Signaling in Eosinophils	2.35	2.75	1.18	3.53	−4.71
GNRH Signaling	2.06	2.95	2.36	3.24	−4.72
NRF2−mediated Oxidative Stress Response	2.89	2.12	2.89	2.89	−3.66
Gαq Signaling	1.72	2.97	2.34	3.90	−3.90
fMLP Signaling in Neutrophils	2.12	3.18	2.12	4.24	−3.54
α−Adrenergic Signaling	2.50	2.05	3.00	3.00	−3.50
VEGF Family Ligand-Receptor Interactions	2.27	2.65	1.89	3.40	−3.78
ErbB Signaling	1.85	2.78	0.93	2.47	−4.94
Macropinocytosis Signaling	2.84	2.40	1.96	3.27	−3.27
Fcγ Receptor-mediated Phagocytosis in Macrophages and Monocytes	2.00	3.33	3.33	2.67	−2.00
Thrombin Signaling	1.64	2.53	2.83	3.13	−3.73
Growth Hormone Signaling	2.61	2.26	1.91	1.57	−3.66
CREB Signaling in Neurons	2.53	2.21	2.85	3.16	−3.16
Tec Kinase Signaling	2.18	2.18	1.91	2.99	−3.27
HGF Signaling	1.94	1.94	1.94	2.53	−4.92
AMPK Signaling	1.67	1.67	2.67	3.33	−3.33
Diseases and Functions					
Organismal death	−4.45	−3.91	−3.8	−4.8	5.41
Cell movement	3.97	4.66	3.21	4.57	−4.62
Cell survival	4.31	5.35	2.80	4.22	−4.63
Migration of cells	3.69	4.88	3.13	4.19	−4.22
Cell viability	4.23	5.12	2.73	4.15	−4.17
Body size	3.41	3.00	4.23	3.74	−3.79
Expression of RNA	2.32	2.68	4.14	3.29	−4.52
Cell proliferation of tumor cell lines	1.57	4.25	2.18	3.50	−3.47
Cell viability of tumor cell lines	3.45	4.34	1.80	3.07	−3.59
Transcription	2.25	2.25	3.95	2.89	−4.54
Cell movement of tumor cell lines	2.62	3.42	2.03	3.63	−3.25
Transcription of RNA	2.30	1.86	3.97	2.75	−4.30
Cell viability of blood cells	1.96	3.39	3.07	3.69	−2.46
Metabolism of protein	1.96	2.50	4.52	3.00	−1.56

The activation z−score is calculated by the IPA software and predicts whether a specific pathway is increased (positive z−score) or decreased (negative z−score) based on the kinase dataset.

## Data Availability

The data presented in this study are available within the article and Appendix A.

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
