# Peer review of "A Metabolomics Investigation of the Metabolic Changes of Raji B Lymphoma Cells Undergoing Apoptosis Induced by Zinc Ions"

_metabolites, 2021, doi:10.3390/metabo11100689_

Round 1

Reviewer 1 Report

The authors analyzed the toxicity of ZnO nanoparticles in cell cultures in vitro. They were especially interested in the evaluation of cytotoxicity, cytoskeletal disturbances and reactive oxygen species production. They evaluated the differences in phosphorylation of 1380 proteins. The metabolomics approach was used to identify the differences in signaling pathways after the treatment of cells with Zinc ions in different time of exposure. The extracted metabolites from the cells after the treatment were analyzed in mass spectrometer. For the protein phosphorylation expression and phosphorylated forms of proteins 1318 antibodies were used in array. The statistical analyses they performed are typical for microarray analysis. They showed the metabolic mechanism of Zn ion – induced apoptosis of Raji lymphoma cells. The increase of metabolites was temporary, higher after 12 hours and diminished by 24 hours. The findings in analysis and the proposed metabolic pathway are interesting.

Author Response

We appreciate generous comments on the manuscript and the effort that you have dedicated to providing your valuable feedback. 

Reviewer 2 Report

This study is to evaluate the metabolomics characteristic and anticancer activity of zinc ions on Raji B lymphoma cells.

Introduction needs to be expansion; some more information should be given: (1) the anticancer activity of zinc ions on lymphoma; (2) the metabolomics characteristic in the apoptosis cells.

Also, it is necessary to point out: the aim of the study is to evaluate the metabolic pathway of Zn-induced apoptosis in Raji B lymphoma cells. (In contrast, ZnO nanoparticles and Zn physiological function are not necessary)

Result: 

(1) Supplementary Figure 1 (Zn ions induced apoptosis of Raji cells via flow cytometry) is an important result, I suggest move it to figure 1.

(2) In Fig 6A, concentration of AA (20 μM) and GPC (20 μM) should also be addressed. In Fig 6B: Figure legend (black and gray bar) should be given in the figure. In Fig 6C: dosage of AA (10 μM) should be given in the figure. In Fig 6D: Y-axis represents: Cleaved PARP/beta actin. Dsage of AA (?μM) should be given.

Discussion:

(1) Line 301~305: More discussion should be given on the role of arachidonic acid (AA) in inducing cancer cell apoptosis.

Author Response

 Thank you very much for reviewing our manuscript. We also greatly appreciate the reviewers for their complimentary comments and suggestions. The manuscript has been revised as suggested by reviewers. Please find attached a point-by-point response to reviewer’s concerns. And the manuscript was edited through the MDPI English Editing service (English editing ID: english-35079).

We hope that you find our responses satisfactory and that the manuscript is now acceptable for publication.
